# Automatic detection, identification and counting of anguilliform fish using *in situ* acoustic camera data: Development of a cross-camera morphological analysis approach

Azénor Le Quinio[1,2]*, Eric De Oliveira[2], Alexandre Girard[3], Jean Guillard[4], Jean-Marc Roussel[1,5], Fabrice Zaoui[2], François Martignac[1,5]

1 UMR DECOD (Ecosystem Dynamics and Sustainability), Institut Agro, IFREMER, INRAE, Rennes, France, 2 EDF R&D LNHE - Laboratoire National d'Hydraulique et Environnement, Chatou, France, 3 EDF R&D PRISME - Performance, Risques Industriels et Surveillance pour la Maintenance et l'Exploitation, Chatou, France, 4 INRAE, CARRTEL, University Savoie Mont Blanc, Thonon-les-Bains, France, 5 Pole MIAME, Management of Diadromous Fish in Their Environment, OFB, INRAE, Institut Agro, University Pau & Pays Adour/E2S UPPA, Rennes, France

* azenor.le-quinio@edf.fr

**Data Availability Statement:** Data recorded on the site of Mauzac are EDF property and are available on request at retd-lnhe-assistantes@edf.fr. The

## Abstract

Acoustic cameras are increasingly used in monitoring studies of diadromous fish populations, even though analyzing them is time-consuming. In complex *in situ* contexts, anguilliform fish may be especially difficult to identify automatically using acoustic camera data because the undulation of their body frequently results in fragmented targets. Our study aimed to develop a method based on a succession of computer vision techniques, in order to automatically detect, identify and count anguilliform fish using data from multiple models of acoustic cameras. Indeed, several models of cameras, owning specific technical characteristics, are used to monitor fish populations, causing major differences in the recorded data shapes and resolutions. The method was applied to two large datasets recorded at two distinct monitoring sites with populations of European eels with different length distributions. The method yielded promising results for large eels, with more than 75% of eels automatically identified successfully using datasets from ARIS and BlueView cameras. However, only 42% of eels shorter than 60 cm were detected, with the best model performances observed for detection ranges of 4–9 m. Although improvements are required to compensate for fish-length limitations, our cross-camera method is promising for automatically detecting and counting large eels in long-term monitoring studies in complex environments.

## Introduction

Active acoustic methods are widely used to study and monitor fish in marine and freshwater ecosystems [1,2]. Acoustic devices emit acoustic waves and record the echoes reflected by

authors confirm that they did not have special access privileges to the data.

**Funding:** The author(s) received no specific funding for this work.

**Competing interests:** The authors have declared that no competing interests exist.

fish that pass nearby. The nature of the echoes depends on fish species and sonar specifications. Acoustic devices thus continuously record data in a non-invasive manner even at night or in turbid water, and at a long detection range (i.e. distance from the device) [2,3]. Many technical improvements have been made over time [4], among which acoustic cameras (i.e. imaging sonar) are key [5,6]. Due to their multiple beams and high frequency, these devices can project the echoes that they detect in a large volume, as 2-dimensional images. Unlike echograms produced by other types of sonar, acoustic cameras can provide high-resolution acoustic videos due to their high rate of emission-reception [5]. Therefore, fish can be visualized as they pass into the acoustic field, during which swimming behavior can be described [5,7] and morphological characteristics such as length can be measured accurately [8,9]. This provides information that is useful for identifying species [10,11]. Recently, in addition to the most commonly used models of acoustic camera–DIDSON and ARIS (Sound Metrics Corp., Bellevue, WA, USA)–new and less expensive models have become available that can scan a larger volume but have lower resolution, such as BlueView (Teledyne Technologies Inc., Thousand Oaks, CA, USA) and Oculus (Ulverston, Cumbria, United Kingdom). Based on these advantages, acoustic cameras are useful for studying long-term migration dynamics in rivers, assessing information on stock and population, and providing new insights for fish-conservation policies [6].

However, continuous recording with acoustic camera produces a large amount of data. Analysis of acoustic video by an operator is a time-consuming process that requires a degree of expertise to distinguish fish from other objects and to identify fish species. Multiple automatic or semi-automatic methods have been developed to detect and describe fish using acoustic camera datasets, listed and described in a recent review [6]. Among the studies quoted, a few authors focused on the distinction of species of interest, such as anguilliform fish, from other species [12–14]. Indeed, an operator can easily distinguish the particular body shape and swimming undulation of anguilliform fish from those of most other fish species [15–18]. Besides, the conservation status of several of anguilliform migratory species, such as the European eel (*Anguilla anguilla* [19]), American eel (*Anguilla rostrata* [20]) and Japanese eel (*Anguilla japonica* [21]), all listed on the IUCN Red List of Threatened Species, make them species of high ecological interest.

Image-processing algorithms, especially computer-vision techniques such as traditional image analysis, machine learning and deep learning, can detect objects automatically in large datasets and extract their morphological characteristics to classify them [22]. Recall and precision rates allow quantifying the ability of those algorithms to identify all target objects (recall) and distinguish them from non-target objects (precision rates). Using videos from acoustic cameras, Bothmann et al. [12] demonstrated the feasibility of using computer-vision techniques to classify European eels automatically according to their body shape and motion, highlighting promising results (recall = 91%; precision = 96%; n = 134 eels). However, the data were recorded using a DIDSON acoustic camera with a short detection range (i.e. 1–6 m), which is rarely used in long-term monitoring studies [7,17,23]. The distance of the fish from the acoustic camera, as well as its body length and orientation may decrease its detectability, because it may lead to the image of its body into distinct disconnected fragments [13,24]. Among recent studies, the deep-learning models of [14] and [25] obtained a recall of 85% when identifying American eels using ARIS sonar recorded under stable flow conditions. The authors recommended that future studies investigate the method's detection ability over long-term monitoring periods. Another recent study that used high-variability training datasets recorded under natural conditions confirmed that eels are among the most difficult fish species to identify using convolutional neural networks, a deep learning method, due to the fragmentation of eel targets in acoustic videos [26].

Based on the results and recommendations of these studies, our objective was to develop a transferrable and new method to detect anguilliform fish automatically from acoustic camera data recorded *in situ* during long-term monitoring surveys, by pairing computer-vision techniques with morphological analysis approaches to correct detection problems caused by fragmentation of fish echoes. To evaluate the effectiveness of the method and identify its potential limitations, we applied it to two datasets recorded in different rivers that had different distributions of silver eel length. The first population is mainly composed by female (body length from 50 to 100 cm [27] although males dominate the second population [28]. Body lengths of European eel male usually range from 35 to 46 cm [27]. In addition, we assessed its effectiveness in being the first cross-camera method, i.e. a method that may automatically detect and identify eels on videos recorded by two models of acoustic cameras, the ARIS and the BlueView, whose resolution and video dimension differ.

## Materials and methods

### Datasets

The datasets were recorded at two monitoring sites. One site (Mauzac, MZC) was located in the Dordogne River (France) at a 50-m wide inlet canal of hydropower plant (S1 Fig). The second site (Port-La-Nouvelle, PLN) was located in the 50-m wide channel between the Bages-Sigean lagoon (France) and the Mediterranean Sea [17]. At both sites, cameras were set perpendicular to the flow with the field of view (FOV) parallel to the river bottom (i.e. scanning horizontally across the channel), covering up to 10 meters from the camera at MZC and up to 14.2 meters at PLN (Table 1). The recorded images represented a top view of the water, with the X and Y dimensions corresponding to the direction of fish movement (upstream or downstream) and the detection range, respectively [5,29]. European eel populations differ between the two sites, with that at MZC containing a large proportion of large individuals (70–90 cm long, according to local fisheries and acoustic datasets) and that at PLN containing mainly small eels (30–60 cm, according to acoustic datasets). More information about the monitoring sites, their eel populations and the sonar settings can be found in [17,30,31].

Two distinct models of acoustic cameras were used to record the datasets: ARIS Explorer 1800 (at MZC and PLN) and BlueView M900-2250-130 2D (at MZC). Recording parameters

**Table 1. Description of the datasets characteristics: Camera model, recording parameters and composition of the datasets, recorded at the two monitoring sites, Mauzac (MZC) and Port-La-Nouvelle (PLN).**

| Dataset | Development | MZC-ARIS | PLN-ARIS | MZC-BV |
|---|---|---|---|---|
| Monitoring site | MZC | | PLN | MZC |
| Acoustic camera | ARIS Explorer 1800 | | ARIS Explorer 1800 | BlueView P900-2250 |
| Frequency (kHz) | 1 800 | | 1 800 | 2 250 |
| Pixel vertical dimension (mm) | 6.8 | | 13.7 | 7.9 |
| Image resolution (px) | 1276×664 | | 926×498 | 1238×2302 |
| Frame rate (frames per sec.) | 7 | | 6 | 5 |
| Field of view width x height (degrees) | 28×14 | | 28×14 | 130×20 |
| Window length limits, from start to stop ranges (m) | 0.7–9.4 | | 1.5–14.2 | 0.2–10.0 |
| Number of eels counted | 24 | 759 | 788 | 198 |
| Eel length (cm) | 70–90 | 70–90 | 30–60 | 70–90 |
| Recording duration (h) | 4.0 | 548.5 | 47.5 | 76.0 |
| Month recorded | Dec. 2018 | Nov. 2014; Dec. 2018 | Nov. 2018 | Dec. 2018 |

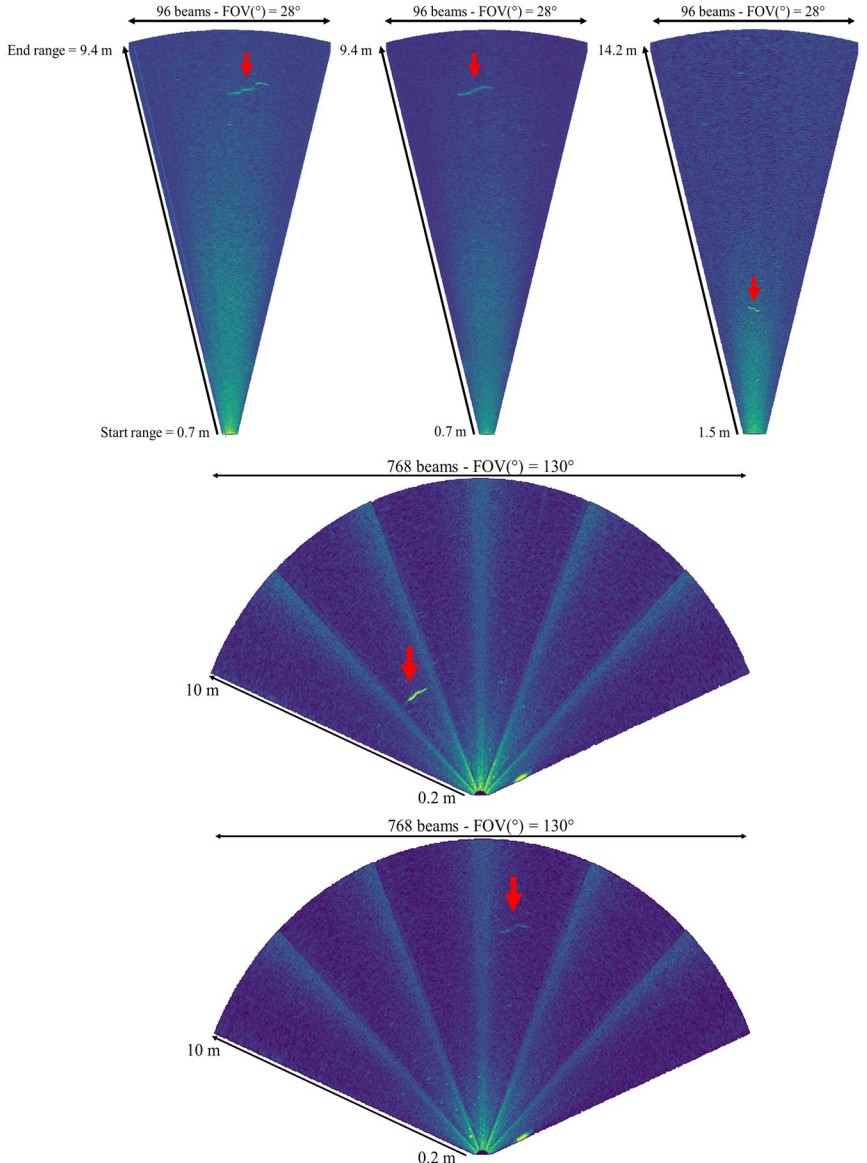

**Fig 1. Example screen captures from ARIS video data recorded at (a, b) Mauzac and (c) Port-La-Nouvelle and (d, e) from BlueView video data recorded at Mauzac.** Red arrows point to eels of different lengths (a: 90 cm; b: 78 cm; c: 30 cm; d: 90 cm; e: 80 cm).

differed between cameras and sites, especially the window length and image resolution (Table 1). All data were recorded in high-frequency mode (1.80 and 2.25 MHz for ARIS and BlueView, respectively) to maximize video resolution and thus eel detection (Fig 1).

## Data processing

All video datasets were watched in their entirety by experienced operators using ARIS Fish software 2.6.3 (ARIS files) or the VLC media player (BlueView files). Each European eel was counted and described by the time that it passed (both MZC and PLN), its detection range (PLN only) and its length, which was measured manually using the most representative frame

chosen by the operator (PLN only, [16]). These visual counts, usually considered the most reliable way to count fish in acoustic camera data [32,33], were used as the reference count (RC) throughout the study. The data were divided into a development dataset (4 h of ARIS data from MZC, with 24 eel passes), one ARIS evaluation dataset from each site (MZC-ARIS, 759 eel passes; PLN-ARIS, 788 eel passes) and one BlueView evaluation dataset from MZC (MZC-BV, 198 eel passes) (Table 1).

## Automatic analysis pipeline

The following method has been developed using Python programming language and especially the opencv [34] and skimage [35] packages.

Our automatic analysis was decomposed in successive steps that used configuration and decision thresholds to carry out the detection, identification, and count of anguilliform fish. Those thresholds were defined empirically from the development dataset. They are adjustable and were based on four metrics: mean pixel intensity $\tilde{I}$ of the reference image (i.e. an empty image without any object passing through), pixel resolution $r$ (mm), the frame rate $fps$ (both imported from the recording settings) and the minimum length of the fish studied $L_{eel\_min}$ (cm), which was set by the operator before the analysis.

**Step 1: Conversion to AVI**. Each raw file of the datasets was converted from its proprietary format (.ARIS or .SON) into a standard video format (.AVI) using a homemade program written in Python.

**Step 2: Detection of the frames of interest**. Intervals of frames of interest were then extracted from the videos, i.e. the frames on which moving discontinuities were passing through the camera FOV. The discontinuities correspond to moving objects that disturb the initial background of the frames. Detection of those frames was carried out from echograms (Fig 2) based on ratios of singular values (SV, [36]) of which detailed calculation is available in S1 File. These echograms were generated by analyzing the video frame-by-frame. The aim was to reduce the method computation time by carrying out the next steps of the analysis only on these targeted frames. They were an alternative to that calculated by the ARIS Fish software, which uses the maximum intensity of echoes [37]. Our use of SV instead of maximums allows us to focus only on elongated shapes, corresponding to anguilliform ones. Hence, our tests on the development dataset videos highlight that the second and third SV explain the main part of

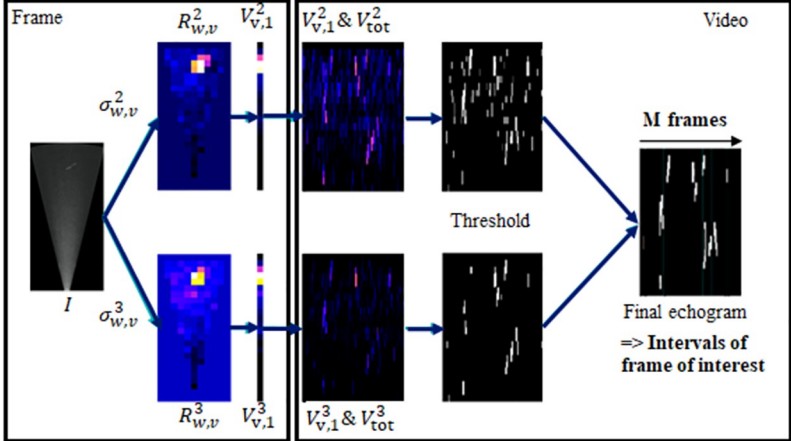

**Fig 2. Workflow of the second step of the method carrying the detection of the intervals of frame of interest from the calculation of echogram based on singular values ratios.**

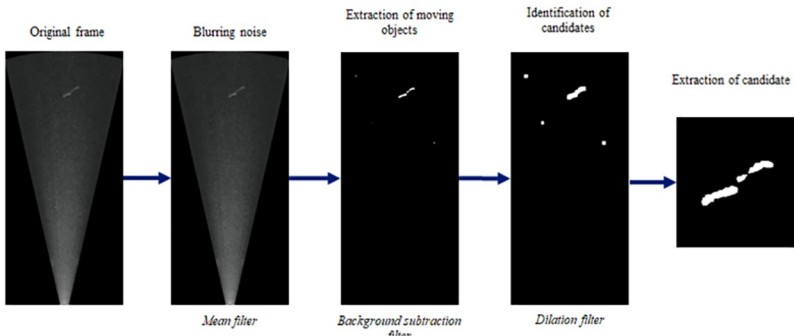

**Fig 3. Workflow of the third step of the method carrying the detection of the candidates.**

the discontinuities. The third SV explains fewer discontinuities than the second one but focuses on the elongated property of the discontinuities. Analyzing the second and third ratios in a complementary way (Fig 2), thus allow us to get the precise intervals of time where elongated discontinuities were passing through and thus to target the corresponding frames.

**Step 3: Detection of the candidates**. Objects corresponding to the discontinuities highlighted at the previous step are extracted from the frames using a combination of image processing filters (Fig 3). A mean filter (kernel = (40 mm/$r$) x (30 mm/$r$)) followed by a Gaussian Mixture-based Background/Foreground Segmentation Algorithm (variance threshold $= 1.5 * \tilde{I}$, history $= 14*fps$, [38,39]) are applied to the frame in order to smooth local variations to reduce noise before isolating moving objects from the background.

A dilation (kernel = 15x15 pixels [40] is then performed on the whole resulting image. The aim is to group the objects that are close to each other making it possible to identify and isolate the regions to be studied. It will prevent an object from appearing on several regions of interest and thus prevent it from being analysed several times.

These successive treatments led to the generation of a binary image of the frame on which the different objects of interest are represented by distinct pixel regions. Objects whose main axis length was less than 25% of $L_{eel\_min}$, i.e. global length of its body main axis which may be different from its body length in case of curved body, were removed because they were considered too short to be an eel. The rest, called "candidates", were individually extracted in a thumbnail (i.e. a square image centered on the object with a dimension of $L_{eel\_min}$ + 30 cm).

**Step 4: Morphological analysis**. In order to classify those candidates as eels or not, each of them was further processed by studying its physical characteristics. Its area (mm$^2$), orientation and overall eccentricity were calculated. The eccentricity is a measure of the non-circularity of the ellipse fitting the candidate shape and is corresponding to the ellipse focal distance divided by the length of its major axis. Its body length was estimated from its "skeleton" image, which was reconstructed for fragmented targets (see S2 File). Because the defining characteristic of anguilliform fish is their serpentine shape, with a uniform body distribution, the shape of its body was quantified by segmenting it using k-means clustering [41]. This method minimized the squared-error function based on the position of body pixels and cluster centroids. The body of the object was divided into three sections (k = 3), as a simplistic representation of the undulation of anguilliform fish with a minimum number of divisions of their body (Fig 4b and 4e). The eccentricity of each section was calculated (Fig 4c and 4f) describing the shape of the body over its entire length.

**Step 5: Tracking from frame to frame**. The candidates were then tracked frame-by-frame during their entire pass through the camera FOV. It was essential to avoid counting the same

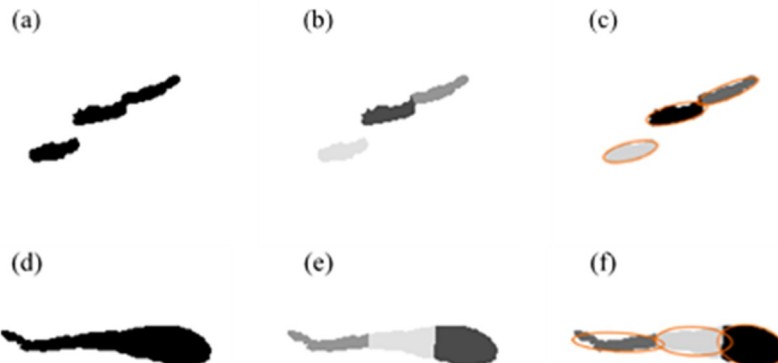

**Fig 4. Steps used to calculate the shape of an echo of an eel (top) or non-eel fish (bottom) pass extracted from an ARIS camera image: (a, d) binary image, (b, e) segmentation using k-means clustering and (c, f) division of the fish body into three sections to calculate eccentricity.**

candidate more than once. Tracking also provided new and valuable information by identifying changes in the physical characteristics of the object in the successive frames in which it was detected, as well as its trajectory (S3 Fig) and velocity, among other information. To handle the tracking, the candidate's neighborhood (circle of diameter 1.5* $L_{eel\_min}$) as well as the consistency of its orientation from one detection to another ($\theta_{diff} < 25°$), were analyzed for a short period after its last detection to determine whether an object that appeared in subsequent frames was the same candidate (S2 Fig).

**Step 6: Classification process**. Finally, candidates were classified as "anguilliform" or "non-anguilliform" based on characteristics calculated throughout the process (Table 2). Each candidate was first classified in each frame before considering its entire track. A candidate was classified as an anguilliform fish if it met all of the decision rules in at least five frames of its track. As for other decision rules, the minimum of five frames was set up from the development database. It corresponds to the trade-off between allowing very short passage to be considered while having enough information to carry out a reliable classification. It could be modified by the user depending on how restrictive he or she wants the classification to be.

All candidates classified as eels were listed in a.CSV output file in the directory that contained the raw dataset and correspond to the automatic counting (AC). Several characteristics

**Table 2. Decision rules for characteristics used to distinguish an eel from another object as a function of the length interval of the eel population.**

| Scale | Characteristic | 30–60 cm | 60–90 cm |
|---|---|---|---|
| Frame | **Length (mm)** | $\geq 0.40 \times L_{eel\_ref}$ | |
| | **Area (mm²)** | $\geq 0.75 \times A_{thumbnail}$ | |
| | **Eccentricity** | $\geq 0.90$ | |
| Track | **Mean travel distance per frame (mm)** | $\geq \frac{L_{eels\_ref}}{fps}$ | |
| | **Velocity (mm/s)** | Track direction was always positive along the axis of the current | |
| | **Eccentricity** | $\geq 0.85$ | $\geq 0.90$ |
| | **Mean eccentricity** | $\geq 0.90$ | $\geq 0.92$ |

$L_{eel\_ref}$, the minimum length of the fish studied, and, $A_{thumbnail}$, the area of the image centered on the object to be classified.

were listed for each predicted anguilliform fish (each line): the video filename, mean time of passage, detection frames, detection ranges, total number of detections and the measured length of one fish per detection frame.

## Evaluation of the effectiveness of the automatic eel counting method

Each video was entirely read by one experienced operator involved in data collection to compare his eels' counting to the predicted eel in the.CSV output file. Operator assessments were consequently used to calculate the number of true positives (TP, a true eel that was predicted correctly), of false positives (FP, a non-eel object incorrectly predicted as an eel) and of false negatives (FN, a true eel that was not detected/predicted). If the automatic method counted the same eel more than once, only the first count was considered a TP, while the others were considered FP, since the method also needed to accurately count the number of eels that passed into the camera FOV. Each FP was categorized to investigate the errors that occurred frequently.

The method was applied to the three evaluation datasets, and its effectiveness and the reliability of its results were assessed by calculating the recall, precision, and F1-score, as well as generating confusion matrices. Recall (Eq 1) highlights the method's ability to identify all eels that passed into the camera FOV. Precision (Eq 2) evaluates the method's ability to distinguish eels from other fish or other objects. The F1-score (Eq 3) summarized recall and precision to assess overall performance of the model. The confusion matrices (Eq 4) determined the TP, FP and FN in the three evaluation datasets. True negatives (i.e. a non-eel object that was predicted correctly) could not be counted, since the operators did not list objects other than eels when reading the videos.

$$Recall = \frac{TP}{TP + FN} \tag{1}$$

$$Precision = \frac{TP}{TP + FP} \tag{2}$$

$$F1\ score = \frac{2 \times Recall \times Precision}{Recall + Precision} \tag{3}$$

$$Confusion\ Matrix = \begin{pmatrix} TP & FN \\ FP & - \end{pmatrix} \tag{4}$$

Linear regressions were calculated to evaluate the relationship between AC and RC and between the number of TP and RC, at an hourly resolution. The slopes of these regressions, their 95% confidence interval and their coefficient of determination ($R^2$) were calculated. For each pair, 70% of the data were bootstrapped 100 times to decrease bias. The slope of the linear regression of each replicate was calculated and Student's t-test was applied to the mean of the 100 slopes to determine whether it differed significantly from 1 (i.e. the slope of the 1:1 line). Differences with $p < 0.05$ were regarded as statistically significant. The two regressions were compared to illustrate the benefit of having an operator validate each prediction to assess the accuracy of the eel count.

Finally, factors that may have influenced the method were evaluated to assess its genericity (i.e. ability to operate well, regardless of the monitoring site and the acoustic camera model). The influence of three factors on the recall, precision and F1-scores were qualitatively studied. The first was the acoustic camera model, for which we analyzed the confusion matrices and

**Table 3. Confusion matrices obtained for the Mauzac site/ARIS camera (MZC-ARIS), Mauzac site/BlueView camera (MZC-BV) and Port-La-Nouvelle site/ARIS camera (PLN-ARIS) datasets.**

| | | MZC | | | | PLN | | |
|---|---|---|---|---|---|---|---|---|
| ARIS | **MZC-ARIS** | *Pred. Eels* | *Pred. Others* | *Total* | **PLN-ARIS** | *Pred. Eels* | *Pred. Others* | *Total* |
| | *Eels* | 556 | 197 | 753 | *Eels* | 349 | 439 | 788 |
| | *Others* | 110 | * | - | *Others* | 485 | * | - |
| | *Total* | 666 | - | - | *Total* | 832 | - | - |
| BV | **MZC-BV** | *Pred. Eels* | *Pred. Others* | *Total* | | | | |
| | *Eels* | 139 | 59 | 198 | | | | |
| | *Others* | 48 | * | - | | | | |
| | *Total* | 187 | - | - | | | | |

Lines are true eels and true other species fish, columns are eels and other species fish predicted by the method * True negatives were not counted.

metrics for a 73-hour period common to the MZC-ARIS and MZC-BV datasets (RC = 56 and 174, respectively). The second factor was the eel's detection range, based on TP and FN distributions, and the change in recall and precision as detection range increased for PLN-ARIS. The third factor was the eel's measured length, based on the relationship between the recall and eel length for PLN-ARIS.

## Results

### Performance of the method

Confusion matrices showed differences between predicted eel detection and the eels observed by operators for the three evaluation datasets (Table 3). MZC datasets had higher F1-scores (72–78%) than the PLN dataset (43%) (Table 4). ARIS and BlueView had similar recall (74% and 70%, respectively), but ARIS had higher precision (84%, vs. 74% for BlueView).

Regression slopes between RC and AC differed significantly (p < 0.001) from 1 for all three datasets, but large positive linear association are observed for MZC-ARIS ($R^2$ = 0.93) and MZC-BV ($R^2$ = 0.95), and weaker for PLN-ARIS ($R^2$ = 0.41), with a slope of 0.47 (Fig 5a–5c). Regression slopes between the number of TP and RC also differed significantly (p < 0.001) from 1 for all three datasets, and the correlations for all datasets were stronger than those between RC and AC but differed more between monitoring sites (Fig 5d–5f). The correlations were strong for PLN ($R^2$ = 0.79), which highlighted a large number of FP at this site and were slightly stronger for MZC-ARIS and MZC-BV ($R^2$ = 0.97 and 0.96, respectively), with similar slopes (0.72) and a small 95% confidence interval (0.01 and 0.02, respectively).

One source of errors for MZC-ARIS and PLN-ARIS (37% and 75% of the FP, respectively) (Table 5) was misidentification due to confusion with other fish species (S4c Fig) or debris. It was the second-largest source of errors (42%) for MZC-BV, while the largest source was tracking errors that resulted in counting the same eel more than once (50%). Other sources of errors

**Table 4. Performance metrics for the Mauzac site/ARIS camera (MZC-ARIS), Mauzac site/BlueView camera (MZC-BV) and Port-La-Nouvelle site/ARIS camera (PLN-ARIS) datasets.**

| Metric | MZC-ARIS | MZC-BV | PLN-ARIS |
|---|---|---|---|
| **Recall** | 73.8% | 70.2% | 44.3% |
| **Precision** | 83.5% | 74.3% | 41.9% |
| **F1-score** | 78.3% | 72.2% | 43.1% |

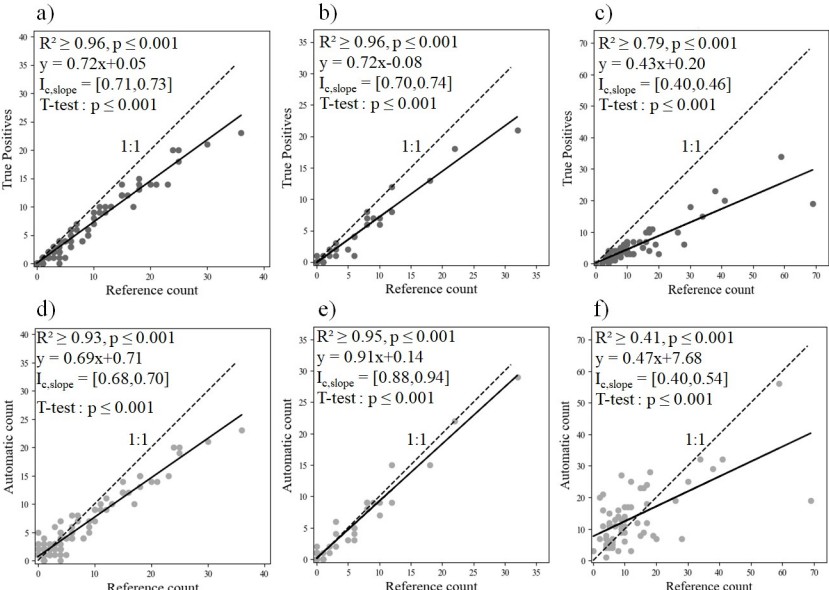

**Fig 5. The method's (a, b, c) true positives and (d, e, f) automatic count of eels as a function of the reference count for (a, d) Mauzac site/ARIS camera (MZC-ARIS), (b, e) Mauzac site/BlueView camera and (c, f) Port-La-Nouvelle site/ARIS camera.** Each plot represents the number of eels counted in one hour (MZC-ARIS: n = 199; MZC-BV: n = 55; PLN-ARIS: n = 60). $I_{c,slope}$ is the 95% confidence interval of the slope.

were arcing effects (S4a Fig), artefacts of high intensity resulting of returning echoes, which deform fish by giving them a more streamlined shape [13], which was observed in MZC-ARIS and PLN-ARIS (36% and 5%, respectively), but not in MZC-BV. Additionally, small fish swimming near each other (S4b Fig) were sometimes detected as one larger individual (20% and 8% of the FP in MZC-ARIS and PLN-ARIS, respectively). Because tracking errors were not considered errors when identifying species, a modified precision (i.e. identification precision) was calculated that excluded them from the FP. Identification precision on MZC-BV was similar to the precision for the ARIS datasets, but was much larger than the precision for identification and counting for MZC-BV (85% for identification and 74% for identification and counting).

## Analysis of factors that influenced performances of the method

The two acoustic camera models at MZC had similar recall and precision (Table 6). For PLN-ARIS, performances peaked 4–9 m from the camera, for which the F1-score exceeded

**Table 5. Percentage of false positives (FP) in each error category observed, total precision and identification precision (i.e. excluding tracking errors) for each dataset: Mauzac site/ARIS camera (MZC-ARIS), Mauzac site/BlueView camera (MZC-BV) and Port-La-Nouvelle site/ARIS camera (PLN-ARIS).** Bold text indicates the largest percentage per dataset.

| | | MZC-ARIS | MZC-BV | PLN-ARIS |
|---|---|---|---|---|
| Error category | Arcing effect | 36.3% (n = 40) | 0% (n = 0) | 4.9% (n = 24) |
| | Merged fish | 20.0% (n = 22) | 8.3% (n = 4) | 3.1% (n = 15) |
| | Other type | **37,3%** (n = 41) | 41.7% (n = 20) | **75.3%** (n = 365) |
| | Tracking | 6.4% (n = 7) | **50%** (n = 24) | 16.7% (n = 81) |
| Total precision | | 83.4% | 74.3% | 41.9% |
| Identification precision | | 84.4% | 85.3% | 44.3% |

**Table 6. Confusion matrices and performance metrics obtained for a common 73-hour period at Mauzac recorded by ARIS and BlueView cameras.**

| ARIS | Pred. Eels | Pred. Others | Total | BlueView | Pred. Eels | Pred. Others | Total |
|---|---|---|---|---|---|---|---|
| Eels | 38 | 18 | 56 | Eels | 122 | 52 | 174 |
| Others | 13 | * | - | Others | 46 | * | - |
| Total | 51 | - | - | Total | 168 | - | - |
| | | Recall | 67.9% | | | Recall | 70.1% |
| | | Precision | 74.5% | | | Precision | 72.6% |
| | | F1-score | 64.8% | | | F1-score | 68.9% |

* True negatives were not counted.

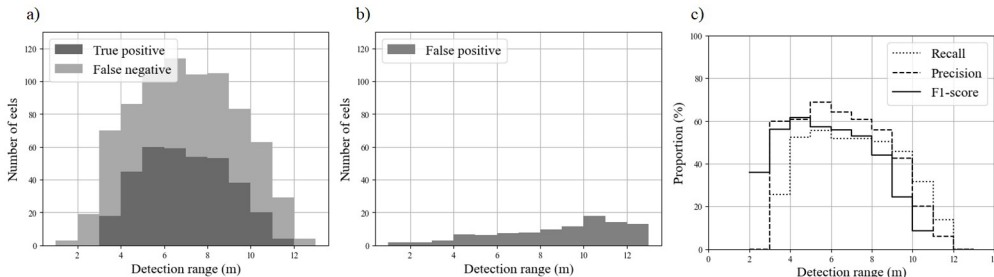

**Fig 6. Performance metrics of the method as a function of the detection range (m) for the PLN-ARIS dataset: a) cumulative true positive and false negative distributions and recall, b) false positive distribution and precision, and c) F1-scores.**

40% (Fig 6a). Beyond 9 m, the precision progressively decreased, with more than 50% of the FP recorded beyond this distance (Fig 6b). Similarly, only 35% of eels that passed more than 10 m and 20% of those that passed within 4 m from the camera were detected (Fig 6c), which resulted in a decrease in recall for these ranges. Eel length also influenced the method's performance. Despite a recall of 44.3% for PLN-ARIS, eel length distributions from TP records and RC were similar (Fig 7). The method's recall decreased below a threshold length of 37 cm but remained near 60% above it. Only one eel larger than the targeted range of 30–60 cm was detected at PLN (78 cm long).

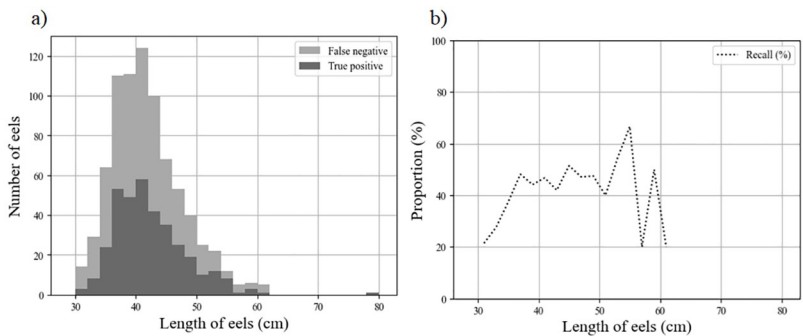

**Fig 7. Performance metrics (cumulative false negative and true positive distributions and recall values) of the method as a function of the measured length of eels (cm) for the PLN-ARIS dataset.**

## Discussion

The method developed to identify anguilliform fish using acoustic camera records performed well compared to those in previous studies [12,14]. The method was able to automatically identify 74% of the large European eels that passed into the ARIS sonar field of view (recall) with 84% precision (i.e. 16% FP). For the same species, Bothmann et al. [12] obtained a higher performance (recall = 91%, precision = 96%) with a DIDSON camera but on a 5m long detection range, *i.e.* 2–3 times shorter than our study. With better image resolution, the fragmented targets were likely not a serious problem for their dataset. Zang et al. [14] also obtained a similar good performance (recall = 84%) on a few individuals of American eels, with no other fish species in their dataset. Moreover, both studies [12,14] used data from a single acoustic camera model at a single monitoring site. The method developed in the present study was tested with two common and different models of acoustic camera and aims to be transferable to other models of acoustic camera. Following the pioneer DIDSON and ARIS cameras, other high-frequency multibeam sonars are now available, notably BlueView and Oculus [42]. These devices are less expensive and compensate their lower resolution by having a wider FOV: they thus cover more than four times as much volume as DIDSON or ARIS. When ARIS was compared to BlueView at MZC, the latter counted three times as many eels during the same recording period. Despite BlueView's lower resolution, the method developed obtained similar recall and precision with it as with ARIS, which highlights the method's genericity. In our study site where both camera types were set up, the BlueView wider FOV increased the capacity of the camera to detect large European eels despite its poorer resolution. Furthermore, our method of counting and detection performed similarly as on ARIS camera for the same population of eel. Previous studies have already shown the potential of the BlueView camera in similar settings [42].

However, more than 50% of the FP recorded in MZC-BV resulted from multiple counting of eels. Considering only the method's ability to distinguish eels from other species or objects, and not the counting accuracy, the precision for MZC-BV was 85.3%, which was similar to that of MZC-ARIS (84.4%). This highlights the method's strong ability to distinguish eels from other objects using either camera model. Because the method also aims to count eels accurately, multiple counts are considered an error, and they decreased the precision for BV-MZC slightly to 74.3%. The larger number of multiple counts in BlueView videos than in ARIS videos was due to the greater difficulty in accurately and continuously tracking the same eel along its path. This error in BlueView videos could have come from two sources: (a) structural noise that appears every 21˚, which prevents capturing an accurate image of objects in the impacted frame quadrants, and (b) the much larger field of view, which makes it more likely to lose an object as it moves and to find it again a little further in the FOV. Post-processing the paths using post-tracking reconstruction to exclude the noise quadrants could help remove duplicates from the eel count.

Despite these frequent errors, the method underestimates the eel count to a similar degree in MZC-ARIS and MZC-BV, with a regression slope of 0.72, a small 95% confidence interval and a strong correlation, despite the differences in the beam formation process of both cameras and in the consecutive resolutions of their images. Those correlations between the number of eels counted by an operator and the eels automatically detected highlight that a constant percentage of eels is efficiently identified by our method on both datasets on the datasets recorded by both camera models even if slopes are significantly different from the 1:1 line. This information could help future studies determine a correction factor site-dependent for the method's count to reach a similar efficiency than an entire reading by an operator.

During their downstream migration, European silver eels display a wide range of body lengths, as well as sexual dimorphism (i.e. males are shorter than females [31]). Overall, sex determination in the European eel is partly related to the local environmental conditions they encounter during their freshwater phase, such as eel density, recruitment and watershed characteristics [43,44]. Our method was more effective for sites dominated by females (recall = 74%, precision = 83%) and performed moderately well when the eels were shorter than 60 cm (recall = 44%, precision = 42%). This limitation was likely due to the camera resolution and fish length. Information about morphology is severely limited when an anguilliform fish is characterized by only a few pixels. Fernandez Garcia et al. [26] obtained similar results with conventional neural networks: their results showed that length had a clear influence on eel recall, decreasing from 65% for eels longer than 60 cm to 22% for those shorter than 60 cm. Despite length-biased performance, we found that the predicted length distribution was consistent with that observed by the operators (Fig 6).

In acoustic camera videos, the horizontal dimension of pixels, or cross-range resolution [6] depends on the detection range, which influences an operator's visualization of fish [45] and may decrease measurement accuracy [46,47]. Our results for the PLN-ARIS dataset highlight the method's robust performances for eels 4–9 m from the camera. Eels closer to the camera were not detected well since their echoes were sometimes hidden or corrupted by arcing effects. These errors are a known issue in DIDSON videos due to fish or highly reflective targets moving near the camera [13,46], but they were not observed in BlueView videos. Mitigating the signal intensity on the recorded videos may decrease the influence of those arcing effects but may also affect the efficiency of automatic fish detections. The specific shape of these artifacts (i.e. the arc of a circle) could be addressed by specific pre-processing of each frame. The ability to identify all eels that passed in the camera FOV peaked 4–10 m from the camera, with a recall of ca. 60%. Similarly, the precision (i.e. the ability to distinguish eels from other objects) also depended on the detection range. Precision peaked 4–9 m from the camera, with a precision of 60%. The number of FP increased greatly beyond 9 m, with errors due mainly to misidentifying other species or debris as eels. This was likely because the increase in the pixel dimension prevented effective identification of small eels. Future studies should focus on whether our model is more effective for detecting and identifying larger eels beyond 9 m.

Unlike the frame resolution of the DIDSON camera, that of ARIS can be set by the operator by adjusting the number of samples along the vertical axis. The vertical resolution can thus range from 3–19 mm, which causes differences in width for fish that move perpendicular to the sonar FOV. Theoretically, we assume that fish identification will improve at a higher resolution. The difference in resolution between MZC and PLN may also have influenced performances at PLN, making it more difficult to distinguish eels from other objects, especially those further from the camera (63% of FP in the PLN dataset). An overly low resolution may also increase the fragmentation of eel echoes that cause discontinuities and the need to rebuild the eel skeleton. In addition to camera recording parameters, future studies should focus on environmental conditions during recording and their influence on identification. Among other factors, water flow can influence the migration activity of silver eels [48] as well as influencing image quality via the number and speed of objects that pass in the camera FOV. Thus, it could have influenced the method's performance by increasing the number of objects detected and shortening the interval of frames of each pass.

## Conclusion

Our study demonstrates the feasibility of using computer vision and morphological analysis to automatically identify large anguilliform fish *in situ*, using two acoustic camera models with

different characteristics, up to a range of 10 m, even when images of fish become fragmented. Unless fish length emerges as a limitation of this method–and future studies should be performed to define its operational boundaries more clearly. Our innovative method can provide relevant ecological data from acoustic cameras during long-term monitoring, in a faster way compared to reading of the entire dataset by a human operator, pushing further the advantage of using acoustic cameras to monitor migratory fish populations without any disturbance on their behavior and health conditions face to more intrusive methods as stow nets.

The results reveal that the automatic method with no post-processing could be useful for monitoring migration dynamics of a large anguilliform fish population. Although it tends to underestimate the total number of fish, it can identify their migration peaks and serve as a proxy of the total number of fish passing through the camera FOV. In a semi-automated application, an experienced operator could validate each predicted anguilliform fish, which would increase the accuracy of the fish count and avoid FP detections. Having a proxy of eel passes is especially useful for the operation of hydropower plants. Future development could include a user-friendly application to monitor *in situ* eels in real time, which could encourage more precise management of turbine shutdowns that optimizes production costs while hurting fewer eels. Besides, the problem of large acoustic data storage can be alleviated by running the analysis right after recording and before archiving the data; the use of higher resolution settings should be much easier too.

Although the method was successful for large silver eels, remaining issues could be investigated to improve it further. Machine learning could be applied to improve the accuracy of the empirically defined thresholds used in the morphological analysis. Fish length, detection range and resolution influence the method's performance. Better understanding the influence of each factor as well as their interactions is a potential area for improvement. Currently, caution is required when using our approach since the automatic detection of small ($< 60$ cm long) eels does not perform very well and counting might be undervalued. Finally, the method was designed and tested only for European eels, but it could be tested for other species with similar undulation and a serpentine body shape, such as the American eel and sea lamprey (*Petromyzon marinus*). Morphological analysis of moving fish is a promising and timesaving approach for using acoustic cameras to identify and count other fish species.

## Supporting information

**S1 Fig. Figure of the installation of the two acoustic cameras at MZC site.**
(TIF)

**S2 Fig. Workflow of the reconstruction process of skeleton in case of fragmented object.** a) Binary image of the candidate's body after background subtraction; b) Binary image of the candidate skeleton; c) Binary image of the candidate reconstructed skeleton.
(TIF)

**S3 Fig. Tracking conditions between two detections of an eel at n frames interval.**
(TIF)

**S4 Fig. Successive positions and shape of an eel along its movement in the camera beam.**
(TIF)

**S5 Fig. Examples of three of the main categories of errors made by the method.** a) Example of an arcing effect; b) Example of the method misidentification due to fish swimming closely to each other, c) Example of the method misidentification as an eel of another fish species.
(TIF)

**S1 File. Calculation of the echograms of singular values ratios.**
(PDF)

**S2 File. Reconstruction and extraction of skeleton characteristics.**
(PDF)

## Acknowledgments

This study was supported by Electricité de France and the French Association Nationale de la Recherche et de la Technologie. We sincerely thank the CEFREM team (University of Perpignan) for the PLN dataset and its visual analysis. We are grateful to the EDF and MIGADO teams who visually analyzed the Mauzac datasets. Finally, we sincerely thank Guy d'Urso for his help and wise advices.

## Author Contributions

**Conceptualization:** Azénor Le Quinio, Eric De Oliveira, François Martignac.

**Data curation:** Azénor Le Quinio, Eric De Oliveira, Alexandre Girard, Fabrice Zaoui.

**Formal analysis:** Azénor Le Quinio, Alexandre Girard, Fabrice Zaoui.

**Funding acquisition:** Eric De Oliveira.

**Investigation:** Azénor Le Quinio.

**Methodology:** Azénor Le Quinio, Alexandre Girard, Fabrice Zaoui.

**Project administration:** Eric De Oliveira, Jean Guillard, Jean-Marc Roussel, François Martignac.

**Software:** Azénor Le Quinio, Alexandre Girard, Fabrice Zaoui.

**Supervision:** Eric De Oliveira, Jean Guillard, Jean-Marc Roussel.

**Validation:** Eric De Oliveira, Jean-Marc Roussel, François Martignac.

**Visualization:** Azénor Le Quinio.

**Writing – original draft:** Azénor Le Quinio, Eric De Oliveira, François Martignac.

**Writing – review & editing:** Azénor Le Quinio, Eric De Oliveira, Alexandre Girard, Jean Guillard, Jean-Marc Roussel, Fabrice Zaoui, François Martignac.

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
