## [Decision Letter · Decision Letter 0]

14 Oct 2022

PONE-D-22-22326Automatic detection, identification and counting of anguilliform fish using in situ acoustic camera data: development of a cross-camera morphological analysis approachPLOS ONE

Dear Dr. Le Quinio,

Thank you for submitting your manuscript to PLOS ONE. After careful consideration, we feel that it has merit but does not fully meet PLOS ONE’s publication criteria as it currently stands. Therefore, we invite you to submit a revised version of the manuscript that addresses the points raised during the review process.

We look forward to receiving your revised manuscript.

Kind regards,

Lorenzo Zane

Academic Editor

PLOS ONE

Journal Requirements:

Additional Editor Comments:

Dear Authors, first of all my apologies for the delay in the review process of your manuscript. It has been extremely difficult for me to secure reviewers.

I now have the thoughtful comments of two reviewers, and both appreciated your manuscript raising some points that need to be addressed.

At this stage my decision is therefore "Major revision", and I ask you to respond point by point at the comments made.

With my best regards.

Lorenzo Zane

Reviewers' comments:

Reviewer's Responses to Questions

**Comments to the Author**

1. Is the manuscript technically sound, and do the data support the conclusions?

Reviewer #1: Partly

Reviewer #2: Yes

2. Has the statistical analysis been performed appropriately and rigorously? 

Reviewer #1: Yes

Reviewer #2: Yes

3. Have the authors made all data underlying the findings in their manuscript fully available?

Reviewer #1: Yes

Reviewer #2: Yes

4. Is the manuscript presented in an intelligible fashion and written in standard English?

Reviewer #1: Yes

Reviewer #2: Yes

5. Review Comments to the Author

Reviewer #1: The MS deals with the Automatic detection, identification and counting of European eels by means of

acoustic cameras deployed in situ.

The Analysis pipeline (fish detection, morphological analysis, frame to frame tracking, classification process) and statistical approach is described in details.

However in my opinion a section of M&M should be dedicated to describe better and even with a figure the Acoustic cameras location and arrangement on the two different sites. The two channels (one in a river and one in a coastal lagoon) are about 50m large and it is quite crucial for the reader to understand how the two devices spatially acted together. It should be noticed that FOV reported for the two acoustic cameras did not exceed 10m. Considering this limitation the applicability of the proposed approach, in addition to poor performance with small-medium size eels (<60cm), might concern more qualitative studies (behavior Activity, migration, depth and spatial distribution, trajectory, swimming direction, swimming speed, tail beat frequency, etc.) rather than quantitative ones. For the sake of scientific honesty such limitation should be highlighted more explicitly, since the target species, the European eel, requires specific monitoring programs in order to collect important data for its sustainable management and stock assessment.

Lastly, this paper is one of the few that presented data obtained by BlueView camera. As the authors state this device is less expensive (an aspect that should not be underestimated in ecological studies) and compensate their lower resolution by having a wider field of view. Some words should be spent to discuss the use of this device in coastal lagoons, not only in rivers, a habitat typology very important in the Mediterranean area. A comparison of the approach used (upgrades and differences) can be done taking into account a similar recent study carried out in Italy with a BLUEVIEW acoustic camera (Capoccioni, F., Leone, C., Pulcini, D., Cecchetti, M., Rossi, A., and Ciccotti, E. (2019). Fish movements and schooling behavior across the tidal channel in a Mediterranean coastal lagoon: An automated approach using acoustic imaging. Fisheries Research 219, 105318. doi:10.1016/j.fishres.2019.105318)

Reviewer #2: Review „Automatic detection, identification and counting of anguilliform fish using in situ acoustic camera data: development of a cross-camera morphological analysis approach”

General comment

In the MS the camera based automatic detection of migrating European eels is tested In doing so, the authors compare two camera set ups. Such automatic approaches could be very useful for a more efficient assessment/monitoring of the annual amount of downstream migrating silver eel compared to more man power consuming approaches like stow nets. Especially as the annual silver eel escapement is used as the indicator for the effectiveness of the implemented European eel regulation (see EC 2007) reliable and cost effective monitoring approaches in the field of eel conservation/monitoring activities.

By having a personal focus on practical eel management related questions, I cannot evaluate the in detail described aspects of the development method and the conducted analysis. Therefore, I hope that the other involved reviewer are more helpful and can provide specific suggestions.

Overall the MS is well written From my perspective, the MS is sometimes too technical. Accordingly, too many details might cause that the broad readership of PLosOne get lost. However, this aspect should be also checked by the editor and the other reviewers. If the other reviews consider this aspect to be less severe, I consider this publishable in PlosOne. Otherwise, I recommend submitting the MS to another journal that has a stronger focus on the technical aspects presented.

At various places in the ms, the citations are given in an “unlucky” style. See for example line 70 “… described in the review of (6). It should be checked if it is possible to provide the names of the authors. If this is not possible, the authors might rephrase these sentences. Check also line 80, 352 and 354.

Specific comments

At various places in the ms, the citations are given in an “unlucky” style. See for example line 70 “… described in the review of (6). It should be checked if it is possible to provide the names of the authors. If this is not possible, the authors might rephrase these sentences. Check also line 80, 352 and 354.

Line 74-76 – to me it is unclear what the authors mean with “high ecological interest” – regarding to what? Migration dynamic? Please specify this statement.

Line 82 – The authors should check, if they can use the terms “recall” and “precision” already in the introduction as the formulas are provided later in the MS (see line 254 and 255)

Line 100 & Line 113-116 – “different distribution of eel length” as well the statement on the sex specific differences in the eel length. I guess what is recorded by the cameras are mostly migrating silver eels on the way to the ocean, which should be clearly stated. Additionally, the given size range for males might be too large. Usually male silver eel have a total length below 50 cm (see Tesch 2003). Against the background of the European eel regulation, this aspect of a clear separation between female and male silver eel should integrated in the MS.

Line 116 – It should be checked if Tesch 2003 could be used to support this statement

Line 130 – How many operators watched the videos. Were there quality controls to unsure a comparable evaluation of the videos?

Line 155 – Maybe I missed, but what means “FOV”?

Line 224 – change into “… how restrictive her or she wants the classification to be.

Line 236 – The information on how many operators were involved should be added.

Line 349-350 – This statement needs supporting references.

Line 356-357 – “Moreover, both studies…” – references need to be added.

Line 349-387 – This is a massive paragraph – I suggest splitting this into 2-3 paragraphes.

Line 389 – Reference 40 should be replaced by a more general one like Tesch 2003

Line 390-391 – Change into “… in the European eel is partly related…” Additionally here the hint, that the European eel represents a facultative catadromous species. Therefore, not every eel is entering the freshwaters during their continental life phase. An unknown proportion remain in saline, brackish or transitional waters.

Line 396 – The way the citation Fernandez Garcia et al. 2021 is given should be checked.

Line 428 – I would add here that the water flow also influence the migration activity of mature silver eels – see for example Reckordt et al. (2014) Ecology of Freshwater Fish

Line 440 – I recommend also here to highlight the advantage of the a camera based monitoring compared to classical silver eel monitoring approaches like stow nets.

Line 453 – for large female silver eels?

References – the given references should be carefuly checked regarding the format style. For example, I guess it is not necessary to provide month of publication for journal articles. Further some references seems to be incomplete like Hughes 2012, Kaifu & Yokouchi 2019 or Dekker & Beaulaton 2016

I recommend to include a map of the sampling sites.

6. PLOS authors have the option to publish the peer review history of their article (what does this mean?). If published, this will include your full peer review and any attached files.

Reviewer #1: No

Reviewer #2: No

---

## [Author Response · Author response to Decision Letter 0]

5 Dec 2022

Reviewer #1: 

The MS deals with the Automatic detection, identification and counting of European eels by means of acoustic cameras deployed in situ.

The Analysis pipeline (fish detection, morphological analysis, frame to frame tracking, classification process) and statistical approach is described in details.

However in my opinion a section of M&M should be dedicated to describe better and even with a figure the Acoustic cameras location and arrangement on the two different sites. The two channels (one in a river and one in a coastal lagoon) are about 50m large and it is quite crucial for the reader to understand how the two devices spatially acted together. 

 Done: a figure of the Mauzac site added to the Supplementary Materials section; a reference to the figure of Lagarde et al. 2021 describing the Port-La-Nouvelle site added line 114.

It should be noticed that FOV reported for the two acoustic cameras did not exceed 10m.: 

 Table 1 summarizes the information on acoustic cameras used in this study. The range of FOV is given in the text (line 116). In Mauzac FOV did not exceed 10 m for both cameras, but in Port-La-Nouvelle, it reached 14 meters. 

Considering this limitation the applicability of the proposed approach, in addition to poor performance with small-medium size eels (<60cm), might concern more qualitative studies (behavior Activity, migration, depth and spatial distribution, trajectory, swimming direction, swimming speed, tail beat frequency, etc.) rather than quantitative ones. For the sake of scientific honesty such limitation should be highlighted more explicitly, since the target species, the European eel, requires specific monitoring programs in order to collect important data for its sustainable management and stock assessment.

 Done, we added a sentence that clearly states this limit (lines 472-474). 

Lastly, this paper is one of the few that presented data obtained by BlueView camera. As the authors state this device is less expensive (an aspect that should not be underestimated in ecological studies) and compensate their lower resolution by having a wider field of view. Some words should be spent to discuss the use of this device in coastal lagoons, not only in rivers, a habitat typology very important in the Mediterranean area. 

A comparison of the approach used (upgrades and differences) can be done taking into account a similar recent study carried out in Italy with a BLUEVIEW acoustic camera (Capoccioni, F., Leone, C., Pulcini, D., Cecchetti, M., Rossi, A., and Ciccotti, E. (2019). Fish movements and schooling behavior across the tidal channel in a Mediterranean coastal lagoon: An automated approach using acoustic imaging. Fisheries Research 219, 105318. doi:10.1016/j.fishres.2019.105318)

 Done, reference to previous study in Mediterranean coastal lagoon by Capoccioni et al. (2019) is given in the text (lines 374-378). 

Reviewer #2: 

Review Automatic detection, identification and counting of anguilliform fish using in situ acoustic camera data: development of a cross-camera morphological analysis approach”

General comment

In the MS the camera based automatic detection of migrating European eels is tested. In doing so, the authors compare two camera set ups. Such automatic approaches could be very useful for a more efficient assessment/monitoring of the annual amount of downstream migrating silver eel compared to more man power consuming approaches like stow nets. Especially as the annual silver eel escapement is used as the indicator for the effectiveness of the implemented European eel regulation (see EC 2007) reliable and cost effective monitoring approaches in the field of eel conservation/monitoring activities.

By having a personal focus on practical eel management related questions, I cannot evaluate the in detail described aspects of the development method and the conducted analysis. Therefore, I hope that the other involved reviewer are more helpful and can provide specific suggestions.

Overall the MS is well written From my perspective, the MS is sometimes too technical. Accordingly, too many details might cause that the broad readership of PLosOne get lost. However, this aspect should be also checked by the editor and the other reviewers. If the other reviews consider this aspect to be less severe, I consider this publishable in PlosOne. Otherwise, I recommend submitting the MS to another journal that has a stronger focus on the technical aspects presented.

At various places in the ms, the citations are given in an “unlucky” style. See for example line 70 “… described in the review of (6). It should be checked if it is possible to provide the names of the authors. If this is not possible, the authors might rephrase these sentences. Check also line 80, 352 and 354. : 

 See below.

Specific comments

At various places in the ms, the citations are given in an “unlucky” style. See for example line 70 “… described in the review of (6). It should be checked if it is possible to provide the names of the authors. If this is not possible, the authors might rephrase these sentences. Check also line 80, 352 and 354. : 

 Done, authors’ names were added whenever needed in the text. 

Line 74-76 – to me it is unclear what the authors mean with “high ecological interest” – regarding to what? Migration dynamic? Please specify this statement.: 

 Done, IUCN Red List of the Threatened Species is now cited (line 76) to highlight the conservation status of the species.

Line 82 – The authors should check, if they can use the terms “recall” and “precision” already in the introduction as the formulas are provided later in the MS (see line 254 and 255) : 

 Done, definition of these terms is now given in the introduction (lines 80-82).

Line 100 & Line 113-116 – “different distribution of eel length” as well the statement on the sex specific differences in the eel length. I guess what is recorded by the cameras are mostly migrating silver eels on the way to the ocean, which should be clearly stated. Additionally, the given size range for males might be too large. Usually male silver eel have a total length below 50 cm (see Tesch 2003). Against the background of the European eel regulation, this aspect of a clear separation between female and male silver eel should integrated in the MS.

 True, our records most likely (but not exclusively) correspond to silver eel. This is clearly stated in the revised version. We also added information and references on sex-dependent body length in eel (lines 103-105). 

Line 116 – It should be checked if Tesch 2003 could be used to support this statement

 Done.

Line 130 – How many operators watched the videos. Were there quality controls to unsure a comparable evaluation of the videos?

 One single operator read videos at Port-La-Nouvelle. At Mauzac, the three operators shared this work; they had a common training in data visualization. The corresponding part of the manuscript (line 243) was rewritten to better explain it.

Line 155 – Maybe I missed, but what means “FOV”?: 

 Done, the term is defined when used for the first time in the text (line 115). 

Line 224 – change into “… how restrictive her or she wants the classification to be.

 Done, line 231.

Line 236 – The information on how many operators were involved should be added.

 Done, see response to previous comment and text (line 243)

Line 349-350 – This statement needs supporting references.: 

 Done, two references added (line 357).

Line 356-357 – “Moreover, both studies…” – references need to be added.: 

 Done, two references added (line 364).

Line 349-387 – This is a massive paragraph – I suggest splitting this into 2-3 paragraphes.: 

 Done, the initial paragraph was separated into 3 parts for clarity (lines 356-401). 

Line 389 – Reference 40 should be replaced by a more general one like Tesch 2003

 Done, line 403. 

Line 390-391 – Change into “… in the European eel is partly related…” :

 Done, line 404. 

Additionally here the hint, that the European eel represents a facultative catadromous species. Therefore, not every eel is entering the freshwaters during their continental life phase. An unknown proportion remain in saline, brackish or transitional waters.

 That is true, but we prefer to stay focused on the freshwater phase in our manuscript. 

Line 396 – The way the citation Fernandez Garcia et al. 2021 is given should be checked. : 

 Done.

Line 428 – I would add here that the water flow also influence the migration activity of mature silver eels – see for example Reckordt et al. (2014) Ecology of Freshwater Fish : 

 True, a mention to water flow is added and the suggested reference is cited (line 442). 

Line 440 – I recommend also here to highlight the advantage of the a camera based monitoring compared to classical silver eel monitoring approaches like stow nets.

 We agreed with the reviewer that the non-intrusive property of acoustic cameras should be highlighted; this is mentioned in the revised manuscript (lines 453-455). 

Line 453 – for large female silver eels? : 

 Done, this clarification has been included (line 468). 

References – the given references should be carefuly checked regarding the format style. For example, I guess it is not necessary to provide month of publication for journal articles. Further some references seems to be incomplete like Hughes 2012, Kaifu & Yokouchi 2019 or Dekker & Beaulaton 2016 : 

 Done, reference list was entirely checked and corrections made whenever necessary.

I recommend to include a map of the sampling sites.

 Done: a figure of the Mauzac site added to the Supplementary Materials section; a reference to the figure of Lagarde et al. 2021 describing the Port-La-Nouvelle site added line 114.

---

## [Decision Letter · Decision Letter 1]

25 Jan 2023

Automatic detection, identification and counting of anguilliform fish using in situ acoustic camera data: development of a cross-camera morphological analysis approach

PONE-D-22-22326R1

Dear Dr. Le Quinio,

We’re pleased to inform you that your manuscript has been judged scientifically suitable for publication and will be formally accepted for publication once it meets all outstanding technical requirements.

Within one week, you’ll receive an e-mail detailing the required amendments. Please also note the last comment of rev2 about the possibile lack of a map. When these have been addressed, you’ll receive a formal acceptance letter and your manuscript will be scheduled for publication.

An invoice for payment will follow shortly after the formal acceptance. To ensure an efficient process, please log into Editorial Manager at http://www.editorialmanager.com/pone/, click the 'Update My Information' link at the top of the page, andy double check that your user information is up-to-date. If you have any billing related questions, please contact our Author Billing department directly at authorbilling@plos.org.

Kind regards,

Lorenzo Zane

Academic Editor

PLOS ONE

Additional Editor Comments (optional):

Reviewers' comments:

Reviewer's Responses to Questions

**Comments to the Author**

1. If the authors have adequately addressed your comments raised in a previous round of review and you feel that this manuscript is now acceptable for publication, you may indicate that here to bypass the “Comments to the Author” section, enter your conflict of interest statement in the “Confidential to Editor” section, and submit your "Accept" recommendation.

Reviewer #1: All comments have been addressed

Reviewer #2: All comments have been addressed

2. Is the manuscript technically sound, and do the data support the conclusions?

Reviewer #1: Yes

Reviewer #2: Yes

3. Has the statistical analysis been performed appropriately and rigorously? 

Reviewer #1: Yes

Reviewer #2: Yes

4. Have the authors made all data underlying the findings in their manuscript fully available?

Reviewer #1: Yes

Reviewer #2: Yes

5. Is the manuscript presented in an intelligible fashion and written in standard English?

Reviewer #1: Yes

Reviewer #2: Yes

6. Review Comments to the Author

Reviewer #1: The authors exhaustively addressed all comments by the two reviewers.

In the present form the paper deserve to be published in PLOSONE

Reviewer #2: Building on the comments of the two reviewers, the manuscript was improved accordingly and is worthy of publication as is. Maybe I missed, but I recommend to include a map showing the both study sites.

7. PLOS authors have the option to publish the peer review history of their article (what does this mean?). If published, this will include your full peer review and any attached files.

Reviewer #1: No

Reviewer #2: No

---

## [Editor Report · Acceptance letter]

16 Feb 2023

PONE-D-22-22326R1 

Automatic detection, identification and counting of anguilliform fish using *in situ* acoustic camera data: development of a cross-camera morphological analysis approach 

Dear Dr. Le Quinio:

I'm pleased to inform you that your manuscript has been deemed suitable for publication in PLOS ONE. Congratulations! Your manuscript is now with our production department. 

Kind regards, 

on behalf of

Dr. Lorenzo Zane 

Academic Editor

PLOS ONE